# Microspherical Titanium-Phosphorus Double Oxide: Hierarchical Structure Development for Sensing Applications

**DOI:** 10.3390/s23020933

**Published:** 2023-01-13

**Authors:** Elena Korina, Anton Abramyan, Oleg Bol’shakov, Vyacheslav V. Avdin, Sladjana Savić, Dragan Manojlović, Vesna Stanković, Dalibor M. Stanković

**Affiliations:** 1Nanotechnology Education and Research Center, South Ural State University, 454080 Chelyabinsk, Russia; 2N.D. Zelinsky Institute of Organic Chemistry, Russian Academy of Sciences, 119991 Moscow, Russia; 3Faculty of Chemistry, University of Belgrade, Studentski trg 12-16, 11000 Belgrade, Serbia; 4Institute of Chemistry, Technology and Metallurgy, National Institute of the Republic of Serbia, University of Belgrade, Njegoševa 12, 11000 Belgrade, Serbia; 5VINČA Institute of Nuclear Sciences-National Institute of the Republic of Serbia, University of Belgrade, Mike Petrovića Alasa 12-14, 11000 Belgrade, Serbia

**Keywords:** hierarchically structured systems, microspheres, water-soluble titanium precursors with organic acids, hydrothermal synthesis, electrochemical sensing

## Abstract

Stable, water-soluble titanium complexed with mandelic acid was used as a precursor for titanium phosphorus double oxide obtained in hydrothermal conditions in the presence of phosphoric acid. Surprisingly, hydrolysis of organic complexes provided a microstructured sphere with narrow size distribution, low aggregation and a small fraction of morphological irregularities. Obtained microspheres had a complex structure comprised of flakes, whose size could be manipulated with temperature conditions. Samples were found to be electrochemically active against sulcotrione, a well-recognized herbicide. Electrochemical sensors based on the synthesized microspheres were successfully adapted for natural water reservoir analysis and exhibited low levels of detection of 0.61 µM, limit of quantification of 1.86 µM, wide dynamic linear range from 2 to 200 µM, good selectivity, excellent reproducibility and in-time stability.

## 1. Introduction

Modern material science has a library of synthetic protocols involving multiple substrates and applying various conditions that yield a plethora of new materials. So, the development of new functional materials is a challenging task that often requires unconventional approaches. Recently, considerable attention was attracted to hierarchically structured materials as a promising division of materials with multiple levels of structural regularity that provide increased functionality. These types of substrates are well recognized: sorbents [1], electrocatalysts [2], electrodes for capacitors [3], photo-anodes [4] and sensors [5], due to their increased specific surface and multimodal porosity. Their multi-regular structure is an excellent prerequisite for utility as a catalyst or as a catalyst substrate. 

Numerous utilizations of hierarchical structures facilitated the widespread development of their formation methods. A major avenue in this area of research is top-down methods based on natural templates [6,7], such as wood [8], leaves and diatoms [9]. In the bottom-up methods, molecular templates, such as surfactants [10], biopolymers [11] and peptides [12] are a commonly used group of structure formation means in the synthesis of sophisticated structures, but their utilization is problematic due to their high cost and low stability. Scarce examples of template-free hierarchical structure formation are still considered exceptions, supporting the general trend for the template-based methodology. In this view, the development of cheap and affordable template-free methods is of high relevance. 

Raising concerns about environmental pollution and toxic impurities facilitates studies in the area of new, prompt and reliable methods for the detection of hazardous materials. Today’s quality of life is secured by the high quality of the food supply, which in turn is attained with complex agricultural measures, including the massive application of herbicides and pesticides for the modulation of crops and cereal harvesting [13]. Sulcotrione is a well-recognized triketone “eco-friendly” herbicide that is used on wide-leafed agricultural plants. However, recent reports on the possible health and environmental effects problematize their use and require systematic monitoring [14,15]. Quality control of herbicide content in distributed food requires prompt, accurate methods for its quantification. Those include dominantly HPLC- and GC-based methods; however, their utilization requires expensive equipment and labor-intensive sample preparation. Other selective and sensitive methods include immunoassay [16] and aptamer-based colorimetry [17], but their selectivity is compromised by the price and fragility of the ligands based on antibodies and aptamers. Electrochemical methods, on the other hand, have become popular in the analytical chemistry of herbicides, and particularly sulcotrione, because they are simple to use, quick and adaptable methods that are widely implemented for the routine control of water and biological samples. 

Major advances in the field of EC-sensors are related to the application of materials with novel micro- and nano-structures with increased surface specificity, signal transfer and selectivity [18]. Recent achievements in the synthesis of nanostructured and hierarchically structured materials gave new momentum to the new electrochemical sensors, enlarging the list of available substrates for analytical application [19,20,21,22,23].

Titanium dioxide is a well-known semiconducting platform used in almost every application field of materials chemistry. It is a robust, non-toxic, abundant and affordable metal oxide that serves as a perfect substrate for new developments in functional materials. One of the major challenges in its synthesis are the precursors, which are often toxic, volatile and pose a significant environmental threat. Moreover, pristine titania is rarely considered for sensing applications: it often requires composite formation or significant surface modification for better performance [24]. Phosphate modification of titania is a promising functionalization method that has been employed for ion-exchange [25,26], catalysis [27,28], electrode formation [29] and photoanodes [30]. We suggested that phosphatization could be adapted for the development of a new electrochemically active composite. To test this hypothesis, we hydrothermally treated a newly synthesized mandelic acid complex of titanium in the presence of phosphoric acid and obtained novel hierarchical structures that were tested in the sensing of the widely used herbicide sulcotrione. The newly synthesized composite exhibited considerably low detection limits, high stability and selectivity and could be suggested for conventional detection of the herbicide. The schematic illustration of the work is given in Figure 1. 

## 2. Materials and Methods

### 2.1. Reagents

DL-Mandelic acid (100%, Bingospa), NH_4_OH (25%, NevaReaktiv), H_2_O_2_ (40%, BiokhimReagent), titanium powder PTM–1, H_3_PO_4_ (98%, Vekton) were used without preliminary purification. 

### 2.2. Electrochemical Measurements

Electrochemical studies were performed at CH Instruments bipotentiostat (CHInstrumetns, Austin, TX, USA), model 760 b. CHI voltammetry software (Version 2.03, CHInstrumetns). A three-electrode voltammetric cell (total volume of 10 mL) was used where unmodified or modified CPE was used as the working electrode, Ag/AgCl as the reference electrode and a platinum wire as the counter electrode. For the electrode characterization, a 5 mM mixture of K_3_[Fe(CN)_6_] /K_4_[Fe(CN)_6_] (1:1) in 0.1 M KCl was used as a testing solution, and electrochemical impedance spectroscopy (EIS) in the range of 10 mHz to 10 kHz and cyclic voltammetry (CV) in the potential range of −0.5 V to 1.2 V at a scan rate of 50 mV/s, were employed. For the development of analytical procedures, square wave voltammetry was used [31]. 

### 2.3. Electrode Preparation 

Unmodified carbon paste electrode (CPE) was prepared by mixing 80% by weight of carbon powder and 20% paraffin oil in a mortar. After a half hour of mixing, a homogeneous carbon paste (CP) was formed. The modified electrodes were prepared by adding a certain amount (by mass) of synthesized materials to the unmodified carbon paste. Percentages were calculated as part of graphite powder only, while the percentage of oil was always constant at 20%. After homogenization of the mixture, the home made Teflon body was filled with carbon paste, and the surface of the electrode was cleaned with paper and used without additional purification [31,32].

### 2.4. Synthesis of Ti-Complex with DL-Mandelic Acid

A total of 0.41 g (8.5 mmol) of titanium powder was dispersed in 5 mL of NH_4_OH(25%), followed by the addition of 40 mL of hydrogen peroxide (37%), to dissolve the metal, controlling the temperature of the mixture within a range of 5–10 °C. The resulting light yellow solution of peroxocomplex was then mixed with a DL-mandelic acid (2.6 g, 17 mmol) solution in 5 mL of water. The resulting water solution is then evaporated at a reduced pressure on a rotary evaporator at 30–40 °C, giving the titanium complex as a light yellow solid. 

### 2.5. Synthesis of Microstructured Titanium-Phosphorus Double Oxide by Hydrothermal Method 

A solution of the titanium complex (0.69 g, 1.66 mmol) in 20 mL of water was placed in a Teflon cup, followed by the addition of 1.15 mL of phosphoric acid (98%). The prepared aqueous solutions was then sealed in a Teflon-coated stainless steel autoclave and heated at 80 and 180 °C for 12 h. Resulting precipitates were decanted and washed with distilled water, followed by centrifugation. Finally, precipitates were dried in the vacuum at 90 °C overnight. Samples obtained at 80 and 180 °C were named AAD66 and AAD85, respectively.

### 2.6. Material Characterization

The registration of IR transmission spectra was carried out on a Shimadzu IRAffinity S1 IR-Fourier spectrometer in the range from 400 to 4000 cm^–1^ with a resolution of 4 cm^–1^ and in 100 repetitions. Elemental analysis and surface morphology of the samples were studied using a Jeol JSM 7001F electron microscope equipped with an Oxford INCA X-max 80 energy dispersive spectrometer. The accelerating voltage of the electron gun was set to the 20 kV required for quantitative EDX analysis. The phase composition and structure of the samples were studied on a Rigaku Optima IV powder diffractometer. The survey was carried out in the range of 2θ angles from 5° to 90° at a survey rate of 5°/min. The study used radiation from a CuKα copper tube (λ = 1.541 Å) at an accelerating voltage of 40 kV. The size distribution of the microspheres was calculated based on 100+ measurements taken using SEM-microphotographs.

## 3. Results

### 3.1. Material Characterization

The majority of the reports about titanium phosphates synthesis rely on the utilization of humidity-sensitive, volatile and toxic titanium derivatives such as titanium tetrachloride [33,34,35,36,37], titanium tetraisopropoxide [38,39,40] or titanium tetrabutoxide [41,42,43]. Phosphatization of titania occurs with phosphoric acid or with water-soluble phosphate salts. In this work, we relied on the previous developments of the Kakihana group [44,45] for the synthesis of water-soluble complexes and used them for the preparation of a titanium dioxide precursor. This approach implies dissolution of metal Ti in hydrogen peroxide followed by organic acid stabilization. Some of the complexes were recrystallized and characterized with single-crystal X-ray [46]. Obtained complexes are non-volatile solids that are stable over long periods of time and could be redissolved and extracted by evaporation from water without loss of a composition. Here, we stabilized Ti-complex with mandelic acid and used it for hydrothermal treatment in the presence of phosphoric acid to obtain a titanium-phosphorus double oxide composite. 

It was found that an equimolar ratio of Ti and P, as well as two- and three-fold excess of phosphoric acid, provided precipitates of amorphous titania with little phosphate content. However, application of a 10+ molar excess of phosphoric acid at 180 °C for 12 h provided a microspherical substrate with a complex structure. SEM microphotographs of the sample are given at Figure 2A. Interestingly, the same morphology of the phosphatized titania was retained at the lower temperature of 80 °C (Figure 2B). Although microstructure formation is not new for the titanium-phosphate double oxide [41,47], the obtained microspheres were a few times bigger in size, had a narrow size distribution (Figure 2D,E), low aggregation and an almost complete absence of morphological irregularities/different shapes. It is evident that microspheres are comprised of distinctive flakes, whose sizes are comparable with those of the spherical aggregate. EDS elemental analysis of microspheres showed even distribution of titanium and phosphorous (Figure 2C,F) with the elemental ratio Ti:P equal to 0.84 for AAD85 and 0.76 for AAD66. Noteworthy is that a freshly prepared Ti-complex without evaporation, as well as a Ti-complex without stabilization with DL-mandelic acid, do not yield microstructure. These two cases demonstrate the negative and positive effects of excessive hydrogen peroxide and mandelic acid on microstructure formation, respectively.

The X-ray diffractogram indicates the obtained materials are amorphous. (Figure 2G) Broad absorption line at 3460 cm^–1^ in the FTIR spectrum (Figure 2H) of the samples corresponds to stretching vibrations of surface hydroxyl groups and adsorbed water molecules. Asymmetric stretching vibrations of NH_4_^+^ and C–H groups are observed at wave numbers 3200 cm^–1^ and 3045 cm^–1^, respectively, in the organic complex and obtained sample [48,49]. The band at 1630 cm^–1^ corresponds to bending vibrations of internal O–H bonds [50,51]. Organic residues of mandelic acid remain embedded in the double oxide microstructure: characteristic bands at 1450 and 1403 cm^–1^ refer to H–C–O scissor and C–O–C asymmetric stretching vibration, while the peaks at 1185 cm^–1^ and 944 cm^–1^ common for all samples are attributed to the C–H bond [49]. Wave numbers 755, 660, 570, and 460 cm^–1^ refer to the stretching vibrations of Ti–O, O–Ti–O, νPO4 and P–O, respectively [52,53,54,55].

### 3.2. Electrocatalytic Properties of the Materials

Electrocatalytic performances of the materials were tested using CV and EIS in a standard probe Fe^2+/3+^ redox couple in order to scrutinize their electron transfer capacities and conductivity. Results are given in Figure 3. In EIS measurements, the resulting plots are composed of two ranges for all the electrodes, where a semicircle is the spectrum at high frequencies and a straight line is the low frequency range. Tested electrodes were bare CPE, AAD66/CPE, where the percentage of AAD material was 10%, and prepared in the same way as AAD85/CPE. The obtained Rct values are contact resistance and can be assigned to the intrinsic resistance of the prepared electrode. These values for all electrodes were calculated as semicircle diameters (Figure 3A). For the tested electrodes, bare CPE, AAD66/CPE and AAD85/CPE, the estimated Rct values were 21,000 Ω, 700 Ω and 1700 Ω, respectively. Compared to other electrodes, AAD66/CPE has the lowest resistance value. These results confirm the assumption that adequately prepared materials influenced the properties of the electrode, including, among other things, the electrical conductivity and wetting of the surface. Among others, it can be concluded that this electrode has the rapidest charge transfer on its surface. To confirm the above-mentioned facts, the electrochemical response of the electrodes was further elucidated using CV measurements in the same working solution (Figure 3B). From these tests, it can be seen that both materials strongly support the diffusion capacities of the bare electrode and increase its electron transfer capability (Figure 3C). However, it is noticeable that the engineering of the material architecture is correlated with the electrochemical properties of the materials. A higher treatment temperature increases the effectiveness of the CPE electrodes in regards to electrochemical parameters. The major morphological difference between the AAD66 and the low-temperature analogue is the well discernible flakes constituting the microspheres. It seems like better-differentiated flakes promote the conductivity in AAD66. The confirmed strong improvement in electron shuttling of the redox probe can promote this material as an efficient candidate for electrode modification and further application for electrochemical detection studies. 

To investigate electrochemical behavior and confirm the diffusion capacities of the materials, both modified electrodes were tested at various scan rates (10 mV/s to 100 mV/s) in the Fe^2+/3+^ redox couple solution. The increase in the scan rate is followed by an increase in the oxidation and reduction currents in the redox system, and these increases can be expressed as a linear dependence of the current from the square root of the scan rate. Both electrodes showed the same behavior. This characteristic behavior for diffusion-controlled processes is additional confirmation of the great potential of the prepared materials for electrocatalytic application. The diffusion-controlled process is further confirmed by plotting log(I) versus log(v) for both electrodes (Figure 3D–G). Finally, the summary of the electrocatalytic examination identifies microspherical titanium-phosphorus double oxide treated at 180 °C (labeled as AAD-66) as a notable candidate for potential application in electrochemical sensorics and further real-world sample analysis.

In addition, to optimize the amount of the modifier in the carbon paste, we employed additional EIS measurements in the iron redox couple with the CPE electrodes with different AAD66 contents—5, 7, 10 and 15%. From the results reviewed in Figure 3H, 10% of AAD66-weighted CPE electrode showed the lowest Rct value, which implies that this amount is most suitable for the preparation of the modified AAD66/CPE. Thus, for the further experiments, we selected this mode of sensor construction.

### 3.3. Electrochemical Performance of Prepared Structures toward Sulcotrione 

The fusibility of the proposed materials toward analyte recognition is a crucial step in the development of electroanalytical methods. For this purpose, we selected CV to inspect the impact of synthesized electrocatalysts on the electrochemical detection of sulcotrione. In Figure 4, the electrochemical oxidation behaviors of the AAD66/CPE and the AAD85/CPE in a 1 mM sulcotrione solution in BRBS at pH 6 were displayed. The operating scan rate was 25 mV/s, and the material content of the CPE was 10%. As can be seen, both electrodes in the selected working potential range provide a well-defined and oval-shaped oxidation peak with a potential maximum at around 1.2 V. No signal was observed in the reversed scan, which indicates irreversible redox behavior of the sulcotrione using the proposed electrocatalysts. These findings clearly confirmed the previously described catalytic properties of these materials, as AAD66/CPE delivers better catalytic properties toward sulcotrione oxidation with a current response of 5.4 µA versus AAD85/CPE with an obtained current of 3.8 µA (Figure 4). This behavior, similar to that of the Fe^2+/3+^ redox couple, can be attributed to the better architecture of the material, which is correlated to the higher active surface area and synergetic effect of titanium and phosphorous. These inferences lead to the final electrocatalytic properties of the materials, such as a high electron transfer rate, higher electrode surface areas and increased diffusion at the interface electrode/testing solution. Based on the conducted study, we can summarize that AAD66 material is suitable for the architecture of titanium/phosphorous oxide nanoparticles for the construction of an electrochemical sensor by modifying the CPE electrode for sulcotrione detection with 10% of the loaded amount of electrocatalyst. 

### 3.4. Effect of Various Parameters toward Sulcotrione Detection Using AAD66/CPE

To better cover the utilization of AAD66/CPE for the electroanalysis of sulcotrione, the effect of varying pH levels was studied using the CV technique in the solution of 1 mM of analyte. Different pHs of BRBS solutions were tested in the range of 2 to 12, with a sweep rate of 25 mV/s (Figure 5A,B). The oxidation current increase was noticeable with the increase of the pH from 2 to 6, as well as with slight changes in the peak potential, which are reflected as a potential decrease toward less positive values (Figure 5C). However, even if this is the case, these potential differences cannot be shown to have a significant correlation with pH. Starting at pH 7, peak current starts to decrease. At pHs higher than eight, residual current leaped rapidly and caused oxidation peak fading (Figure 5B). As a summary, it should be emphatically stated that the highest sulcotrione current response was obtained at pH 6 on AAD66/CPE. Thus, the electrochemical studies in the research were conducted in this supporting electrolyte.

CV, at different sweep rates (5–200 mV/s), accomplished over AAD66/CPE in sulcotrione solution, was used to elucidate the nature of the interface reaction (Figure 5D). With the scanning rate rising, a rising value of the peak current was noticed (Figure 5E). This increase was continuous and can be expressed as I_a_ vs. the square root of the scan rate, where I_a_ is the oxidation peak current. The regression equation for this dependence is I_a_ = 0.8723 v^1/2^ (mv/s)^1/2^ – 0.1824, with a regression coefficient of 0.990, indicating the diffusion-controlled nature of the electrochemical reaction of sulcotrione oxidation over AAD66/CPE. To further confirm this, the dependence of log(I_a_) vs. log(v) was calculated (Figure 5F). The regression equation of log(I_a_) = 0.5043 log(v) – 0.0812 yields a linear response with an R^2^ = 0.993. The slope of this linearity was almost equal to the ideal value of 50 mV, further confirming the dominant diffusion-controlled nature of the interface reaction. 

### 3.5. Sulcotrione Determination on AAD66/CPE 

#### 3.5.1. Method Selection and Optimization 

To properly offer electroanalytical methods for targeted analyte determination, the most common pulse methods are tested. In our study, we compare the differential pulse voltammetric (DPV) and SWV methods toward sulcotrione detection over AAD66/CPE in BRBS at pH 6. Results are sectioned in Figure 6. Both techniques provide electrochemical oxidation of the analyte; however, using the SWV method, a notably higher oxidation peak current is obtained. Hence, we can select this technique as a more appropriate candidate for the development of the analytical procedure for sulcotrione determination. Further studies were conducted with the aim of optimizing SWV parameters, which can effectively contribute to the final analytical characteristics of the procedure. Pulse amplitude (ranging from 10 to 100 mV), frequency (ranging from 10 to 100 Hz) and increment (ranging from 2 to 16 mV) were tested. During the optimization of one parameter, the others were kept constant. In such a way, we acquired the following values: pulse amplitude—20 mV, pulse increment—10 mV and frequency—40 Hz, as optimal, and we used them for procedure evolution. 

#### 3.5.2. Analytical Method Development 

As the most important outcome of this study, we measured the electrochemical response of the AAD66/CPE electrode with varying sulcotrione concentrations. All experiments were conducted under previously optimized experimental conditions, and voltammograms were recorded with the SWV technique. The results are shown in Figure 7A, while the resulting calibration plot is provided in Figure 7B. From these results, we can see that the rise in the sulcotrione concentration is accompanied by an increase in the resulting current response. This increase shows a linear response of current vs. concentration in a very wide range from 2 to 200 µM with the following relationship: I_a_ (µA) = 0.0064 × c (µM) + 0.0043 and a regression coefficient of 0.989. LOD, calculated as 3σ/slope, of the proposed method was calculated to be 0.61 µM, and LOQ, calculated as 10σ/slope, of 1.86 µM. These performances can be attributed to the structural architecture of the material and evaluated electrocatalytic properties, transfer kinetic and diffusion rate. The results, in terms of limit of detection and linear range, are comparable with the recently published studies. Due to the lack of data about this topic, we can conclude that our results are more or less similar to the results obtained by Stanković et al. and Rocaboy-Faquet et al., while Rajiji and coworkers provided a lower LOD and a wider linear range [56,57,58]. The repeatability of the method was tested with the five measurements of the two concentrations of sulcotrione, 20 and 80 µM (Figure 7C,D). At both concentration levels, the relative standard deviations of the ten measurements were lower than 4.5%, indicating excellent accuracy and precision of the developed approach. Stability studies were conducted over one month. During this time, electrodes were stored under laboratory conditions. By testing 20 µM of sulcotrione solution every four to five days during this period (Figure 7D), the obtained current had an RSD of 4.1%, which is in the range of RSD for reproducibility measurements, which indicates that during this period, the current did not change significantly and that the electrode retained its original properties.

### 3.6. Interference Studies and Practical Applicability 

Selectivity is a critical component for efficient electrochemical sensors. With the aim to investigate selectivity for the AAD66/CPE sensor, we tested several species that can be found during practical application of the method (Figure 8). In the presence of common and widely present ions, oxidation current was unchanged, except for the nitrite and fluoride ions, where obtained current rose dramatically. However, nitrite and fluoride ions, especially in the tested concentration, are present in the specific waters, and we can still conclude that our method can be used in common practice. In the presence of interesting organic compounds, the proposed sensor shows good selectivity. Significant current changes were noted only in the presence of the structurally similar pesticide, mesotrione. As these two pesticides do not have common use in the field, we can assume that their joint presence in water is unexpected. Based on previously conducted studies, we can conclude that the proposed sensor possesses satisfactory selectivity and can be used for practical applications. 

Practical applicability was tested with the utilization of the AAD66/CPE sensor for the determination of sulcotrione levels in real-time water samples. Two types of samples were selected: pipe water and wastewater. Samples were analyzed according to the developed procedure. Sampling was as follows: samples were taken directly from the source and stored in the refrigerator until the day of testing. On the day of work, samples were diluted with BRBS pH 6 in a ratio of 1:1 and tested directly. After investigation of the samples, recovery studies were conducted with the addition of standard sulcotrione solution. The spiked amounts were 2, 4 and 5 µM. Resulting voltammograms are summarized in Figure 9, while results obtained from the calibration curve are given in Table 1. Recovery outcomes from real samples were comparable with the added amount of sulcotrione, with results fluctuating from 95 to 103%, and these results prove that the proposed sensor can be successfully applied for sulcotrione monitoring in related samples. 

## 4. Conclusions

In this work, microspherical titanium-phosphorus double oxide nanostructures were synthesized, and morphological tests with XRD and SEM confirmed crystal structure, uniformity and flake architecture. Both materials serve as excellent current amplifiers for the bare carbon paste electrode. The nanohybrid structure obtained at a higher treatment temperature possesses a better conductive nature and served as an excellent platform for the development of an electrochemical sensor with an application in the analysis of the environmental contaminant sulcotrione. Excellent analytical parameters were obtained in terms of limit of detection, linear working range and selectivity. The applicability to real-time samples demonstrated in this work suggests that this sensor exhibits great potential for further studies and possible extension from laboratory application to commercial use. 

## Figures and Tables

**Figure 1 sensors-23-00933-f001:**
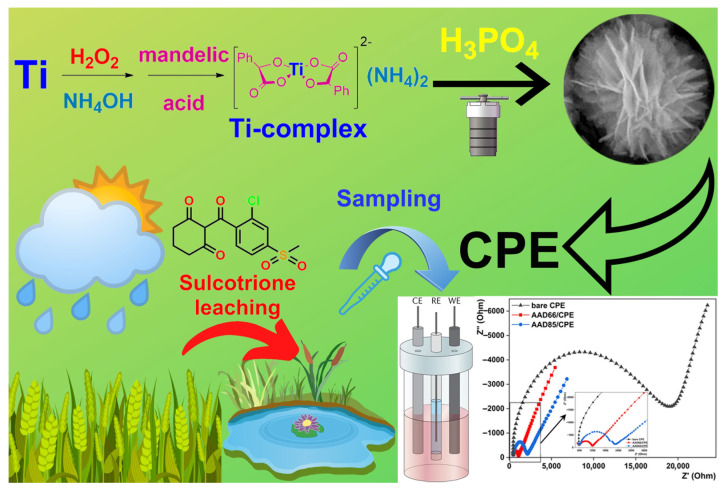
Preparation route for idea AAD microstructures and illustration of the work.

**Figure 2 sensors-23-00933-f002:**
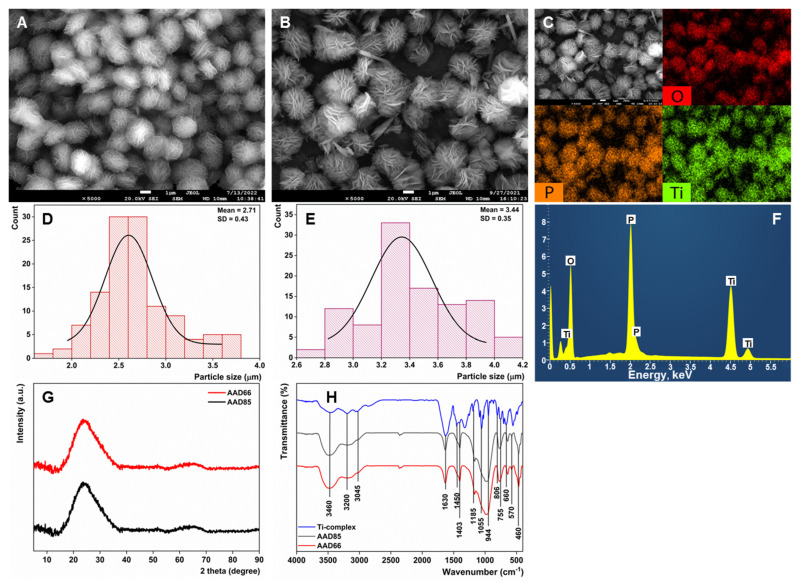
(**A**) SEM-microphotographs of AAD85; (**B**) SEM-microphotographs of AAD66; (**C**) EDS-elemental mapping of AAD66; (**D**) microsphere size distribution of AAD85; (**E**) microsphere size distribution of AAD66; (**F**) EDS-spectra of AAD66; (**G**) X-ray diffractometry of AAD samples; (**H**) FTIR-spectra of the AAD-samples.

**Figure 3 sensors-23-00933-f003:**
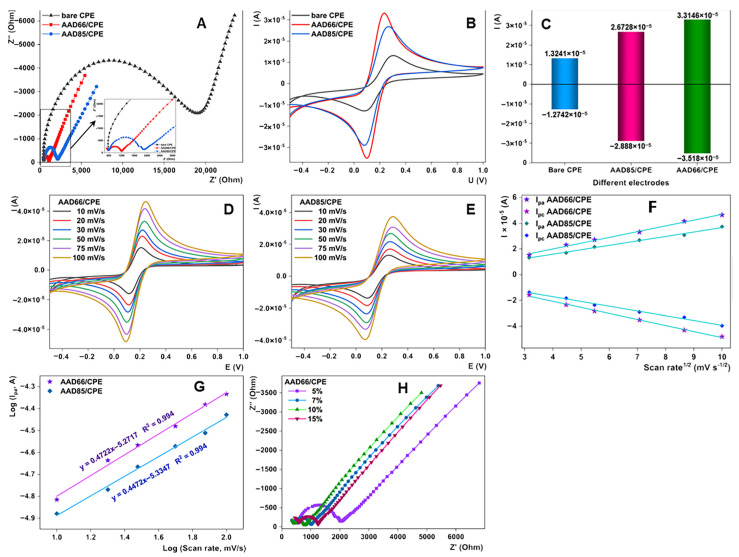
(**A**) EIS spectra for unmodified CPE, AAD66/CPE and AAD85/CPE modified electrodes; (**B**) CV curves in the Fe^2+/3+^ redox couple for unmodified CPE, AAD66/CPE and AAD85/CPE modified electrodes; (**C**) Current intensities for unmodified CPE, AAD66/CPE and AAD85/CPE modified electrodes; (**D**) CVs in the Fe^2+/3+^ redox couple for AAD66/CPE at various scan rates (10–100 mV/s); (**E**) CVs in the Fe^2+/3+^ redox couple for AAD85/CPE at various scan rates (10–100 mV/s); (**F**) Dependence from Ia and Ic for AAD66/CPE and AAD85/CPE vs. v^1/2^; (**G**) Dependence from log(Ia) vs. log(v) for AAD66/CPE and AAD85/CPE; (**H**) EIS spectra for different percentages (5, 7, 10 and 15%) for AAD66/CPE.

**Figure 4 sensors-23-00933-f004:**
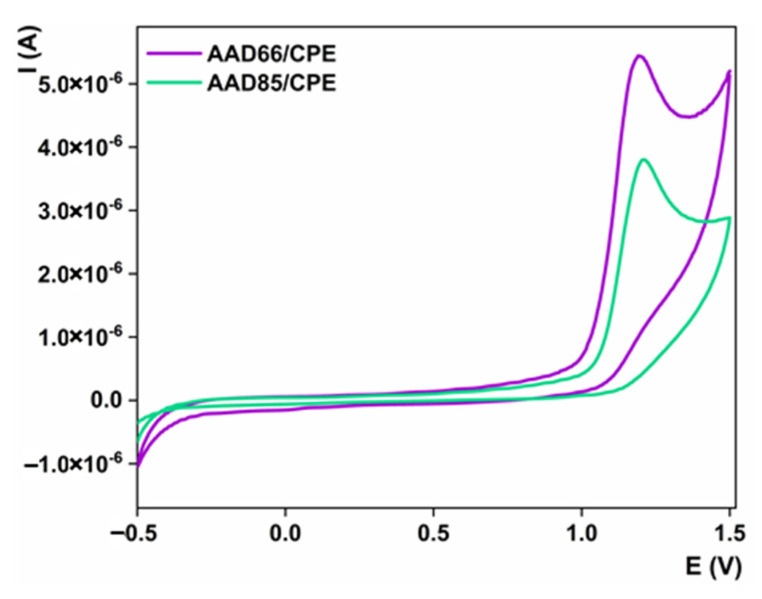
Electrochemical profile of 1 mM of sulcotrione over AAD66/CPE and AAD85/CPE; Current intensities for 1 mM 1 mM of sulcotrione over AAD66/CPE and AAD85/CPE. Operating conditions: scan rate of 25 mV/s; supporting electrolyte BRBS pH 6.

**Figure 5 sensors-23-00933-f005:**
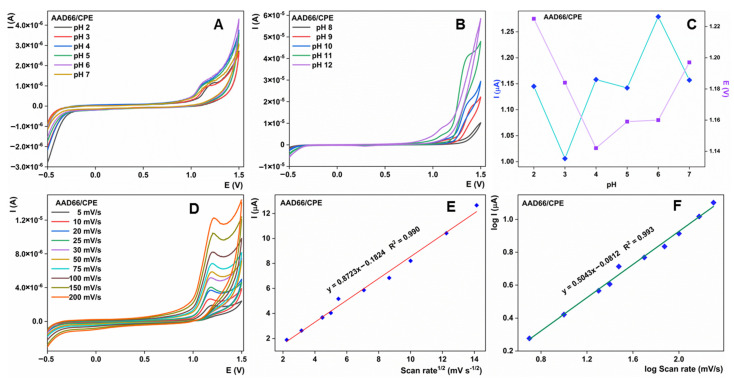
CVs at 25 mV/s for 1 mM of sulcotrione at AAD99/CPE in the ranges of (**A**) 2–7; (**B**) 8–12; (**C**) Dependence of I_a_ and E_a_ vs. pH; (**D**) CVs at various scan rates (5–200 mV/s) for 1 mM of sulcotrione in BRBS pH 6; (**E**) Dependence of the I_a_ from the square root of the scan rate; (**F**) Dependence of log(I_a_) vs. log(v).

**Figure 6 sensors-23-00933-f006:**
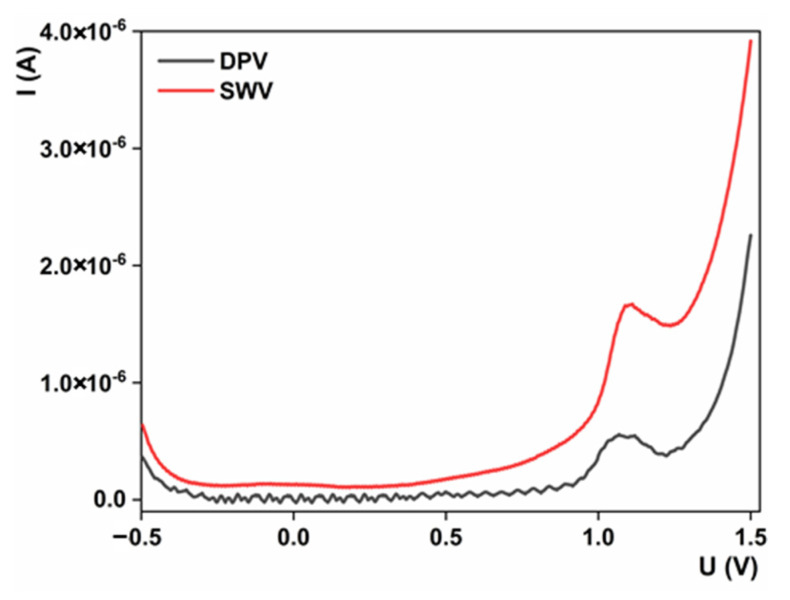
SWV and DPV responses for 1 mM of sulcotrione over AAD66/CPE in BRBS pH 6; Current intensities for both techniques.

**Figure 7 sensors-23-00933-f007:**
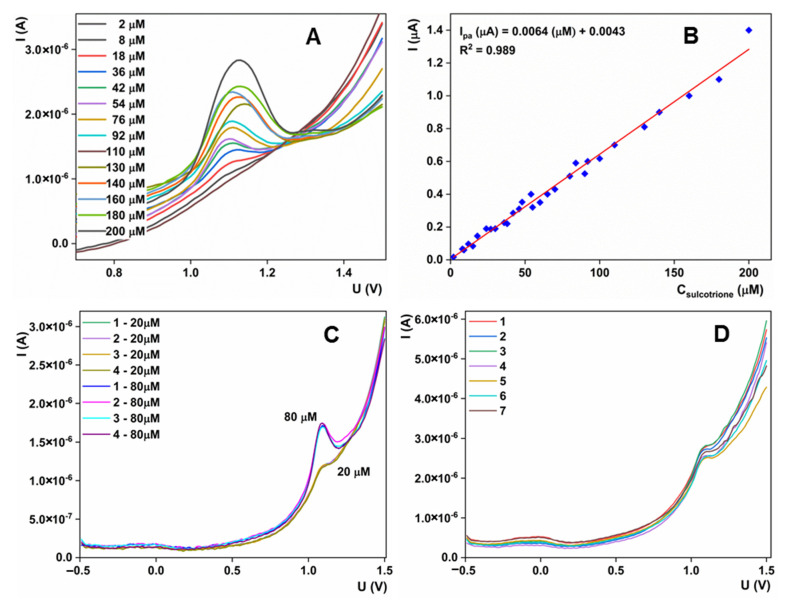
(**A**) SWV voltammograms obtained for different sulcotrione concentrations under optimized experimental conditions; (**B**) Resulting calibration plot; (**C**) Repeatability studies; (**D**) Stability studies.

**Figure 8 sensors-23-00933-f008:**
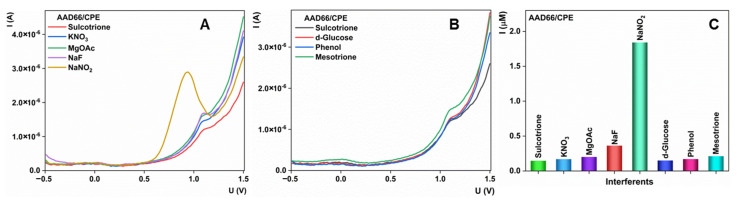
(**A**) Effect of selection ions on the determination of sulcotrione with the AAD66/CPE sensor; (**B**) Effect of organic compounds on the determination of sulcotrione with the AAD66/CPE sensor; (**C**) Current values for sulcotrione with and without interferences.

**Figure 9 sensors-23-00933-f009:**
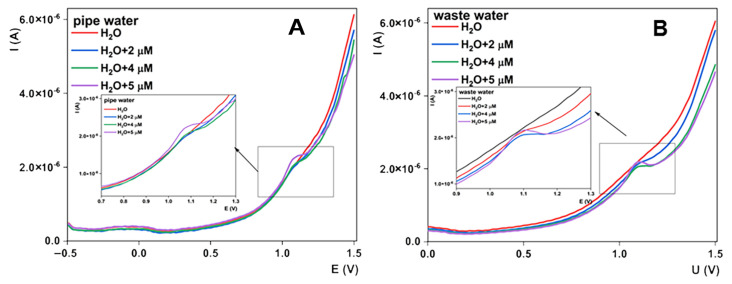
Detection and recovery studies for sulcotrione in (**A**) pipe water sample and (**B**) wastewater sample.

**Table 1 sensors-23-00933-t001:** Results for sulcotrione detections and recovery studies in real-time samples.

	Found (µM)	Added (µM)	Found (µM)/Recovery (%)	Added (µM)	Found (µM)/Recovery (%)	Added (µM)	Found (µM)/Recovery (%)
Pipe water	0.00	2.00	1.94/97	2.00	4.04/101	1.00	5.13/103
Wastewater	0.00	2.00	1.98/98	2.00	4.09/102	1.00	5.08/102

## Data Availability

The study did not report any data.

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
