# Peer review of "Microspherical Titanium-Phosphorus Double Oxide: Hierarchical Structure Development for Sensing Applications"

_sensors, 2023, doi:10.3390/s23020933_

Round 1
Reviewer 1 Report
Figure 1 should be edited to contain only information relevant to the information in chapter 3.1. It looks like graphical abstract in its current state.
Figure 4B could be deleted, the required information about the values of peak currents is already in the text. The same applies also to Figure 6B.
Signals in voltammograms in Figure 9 should be magnified to see the actual response of the analyte in both samples.
The name of the analyte must be unified. There are names “sulcotrione”, “sulcotrone”, and “sulctrione”, all for one compound.
Authors should again proofread their manuscript. Grammar and style in Abstract – “Samples were found…”. How “samples” can be electrochemically active? The degree of Celsius should be written as “°C”, and the reciprocal wavelength as “cm–1”. Oxidation states in Fe should be written with superscript, etc.
The authors mentioned “Excellent analytical parameters were obtained in terms of limit of detection, linear working range and selectivity.” A comparison with already published work should be presented to support this statement in the conclusion, especially when the corresponding author published another work on this topic in the past.
Minor revision of the manuscript is required.
Author Response
Dear Editor,
Thank you so much for giving us an opportunity to revise our manuscript, and thus improve our work according to the reviewers’ valuable comments. We have carefully considered all reviewers’ questions and all the weak points of our work and have significantly changed our manuscript accordingly. The list of changes has been written below, and inserted changes in the text of the re-submitted manuscript are being highlighted. We hope that the revised version of the manuscript will satisfy your and the reviewers’ requirements for publication in the Journal.
Reviewers comments:
Reviewer 1.
We are grateful to the reviewer for its time and efforts and positive comments of our work. We revised manuscript according to these comments and we hope that this improvement, based on reviewer comments will significantly improve final version of our work.
Figure 1 should be edited to contain only information relevant to the information in chapter 3.1. It looks like graphical abstract in its current state.
Thank you for your suggestion. We agree with the reviewer that the current place of the Figure 1 is not appropriate and in the revised version we moved Figure 1 after Introduction section in order to illustrate schematically the whole idea of the work.
Figure 4B could be deleted, the required information about the values of peak currents is already in the text. The same applies also to Figure 6B.
Thank you for commenting this. In the revised version we accept this and revised Figure 4 and Figure 6.
Signals in voltammograms in Figure 9 should be magnified to see the actual response of the analyte in both samples.
Thank you for your suggestion. In the revised version we improved Figure 9 and magnified response of the analyte. In the revised version new Figure 9 is provided.
The name of the analyte must be unified. There are names “sulcotrione”, “sulcotrone”, and “sulctrione”, all for one compound.
Thank you for this comment. This was technical mistake. In the revised version we omitted all wrong names.
Authors should again proofread their manuscript. Grammar and style in Abstract – “Samples were found…”. How “samples” can be electrochemically active? The degree of Celsius should be written as “°C”, and the reciprocal wavelength as “cm–1”. Oxidation states in Fe should be written with superscript, etc.
Thank you very much for this comment. This is an technical mistake. In the revised version we omitted all these mistakes.
The authors mentioned “Excellent analytical parameters were obtained in terms of limit of detection, linear working range and selectivity.” A comparison with already published work should be presented to support this statement in the conclusion, especially when the corresponding author published another work on this topic in the past.
Thank you for this suggestion. In the revised version we provided a discussion about recently found studies. According to our best we found three different studies and provided corresponding discussion in the section 3.5.1 Method selection and optimization. This is highlighted in the revised version.
Minor revision of the manuscript is required.
Reviewer 2
In this paper microspherical titanium-phosphorus double oxide nanostructures were synthesized and morphological tests with XRD and SEM confirmed crystal structure, uniformity and flakes architecture. Both materials serve as current amplifiers for bare carbon paste electrode. The nanohybrid structure obtained at higher treatment temperature possesses better conductive nature and served as platform for the development of electrochemical sensor with the application in analysis of environment contaminant sulcotrione. Analytical parameters were obtained in terms of limit of detection, linear working range and selectivity. The applicability in real-time samples demonstrated in this work suggests that this sensor exhibit potential for further studies and possible extension use.
Thank you for your time and effort. We appreciate all comments and conclusions. In the revised version, we have accepted all valuable suggestions and tried to improve our manuscript. All changes are highlighted in the revised version.
There are many problems need be modified in superscript and subscript, eg,
superscript problem, cm-1 in line 129, 183, 184, 186, 188, NH4+ in line 182, Fe2+/3+ in line 192, 219, 239, 241, 242, 260, Pb2+, Zn2+, Gd2+ in line 498.
subscript problem, Nh4OH, H2O2, H3PO4 in line 88-89, K3[Fe(CN)]6/K4[Fe(CN)]6 in line 97, PO4 in line 189, Ti(HPO4)2·H2O in line 488, TiO2 in line 510.
We are very grateful to the reviewer for noticing this. In the revised version we omitted all technical mistakes.
There many problems need be modified in figure.
It is not preparation route for inset figure at lower right of Figure 1, it is EIS spectra for different parameters. It is not EDS-spectra of AAD66 for Figure 2(e), it is microsphere size distribution,By the way, should describe the figure 2(f) in the paper.
Thank you for noticing this. These are technical mistakes. Our idea was to schematically illustrate idea of the work. We rearranged this. In the revised version we moved scheme 1 after Introduction section and labeled this scheme correctly: Figure 1. Preparation route for idea AAD microstructures and illustration of the work.
Thank you for the comments for the Figure 2. In the revised version we did all the required changes and omit our mistakes.
Reference problems,
Reference 33 should be described in English, first name and family name are wrong for authors in reference 37.
Thank you for this remark. In the revised version we did these suggestions.
XR need change to XRD in line 387.
Thank you for this comment. Authors mistake. In the revised version this is done.

Reviewer 2 Report
In this paper microspherical titanium-phosphorus double oxide nanostructures were synthesized and morphological tests with XRD and SEM confirmed crystal structure, uniformity and flakes architecture. Both materials serve as current amplifiers for bare carbon paste electrode. The nanohybrid structure obtained at higher treatment temperature possesses better conductive nature and served as platform for the development of electrochemical sensor with the application in analysis of environment contaminant sulcotrione. Analytical parameters were obtained in terms of limit of detection, linear working range and selectivity. The applicability in real-time samples demonstrated in this work suggests that this sensor exhibit potential for further studies and possible extension use.
There are many problems need be modified in superscript and subscript, eg,
superscript problem, cm-1 in line 129, 183, 184, 186, 188, NH4+ in line 182, Fe2+/3+ in line 192, 219, 239, 241, 242, 260, Pb2+, Zn2+, Gd2+ in line 498.
subscript problem, Nh4OH, H2O2, H3PO4 in line 88-89, K3[Fe(CN)]6/K4[Fe(CN)]6 in line 97, PO4 in line 189, Ti(HPO4)2·H2O in line 488, TiO2 in line 510.
There many problems need be modified in figure.
It is not preparation route for inset figure at lower right of Figure 1, it is EIS spectra for different parameters. It is not EDS-spectra of AAD66 for Figure 2(e), it is microsphere size distribution,By the way, should describe the figure 2(f) in the paper.
Reference problems,
Reference 33 should be described in English, first name and family name are wrong for authors in reference 37.
XR need change to XRD in line 387.
Author Response

(The authors gave the same response as above.)
